# Waiting Time between Breast Cancer Diagnosis and Treatment in Brazilian Women: An Analysis of Cases from 1998 to 2012

**DOI:** 10.3390/ijerph17114030

**Published:** 2020-06-05

**Authors:** Naidhia Alves Soares Ferreira, Jean Henri Maselli Schoueri, Isabel Cristina Esposito Sorpreso, Fernando Adami, Francisco Winter dos Santos Figueiredo

**Affiliations:** 1Laboratory of Epidemiology and Data Analysis, University Center Health ABC - FMABC, Santo André 09060-870, Brazil; naidhiasoares@hotmail.com (N.A.S.F.); jean.schoueri@gmail.com (J.H.M.S.); adamifernando@uol.com.br (F.A.); 2Oncológica do Brasil Ensino e Pesquisa, Research Center, Belém 63053-000, Brazil; 3University Center of Juazeiro do Norte—UNIJUAZEIRO, Juazeiro do Norte 63010-475, Brazil; 4Faculty of Medicine, University of São Paulo, São Paulo 05403-000, Brazil; icesorpreso@usp.br

**Keywords:** breast cancer, epidemiology, unified health system, public policies, education levels and cancer, socioeconomic status and cancer

## Abstract

Brazilian law requires that treatment for breast cancer begin within 60 days of diagnosis. This waiting time is an indicator of accessibility to health services. The aim of this study was to analyze which factors are associated with waiting times between diagnosis and treatment of breast cancer in women in Brazil between 1998 and 2012. Information from Brazilian women diagnosed with breast cancer between 1998 and 2012 was collected through the Hospital Registry of Cancer (HRC), developed by the National Cancer Institute (INCA). We performed a secondary data analysis, and found that the majority of women (81.3%) waited for ≤60 days to start treatment after being diagnosed. Those referred by the public health system, aged ≥50 years, of nonwhite race, diagnosed at stage I or II, and with low levels of education waited longer for treatment to start. We observed that only 18.7% experienced a delay in starting treatment, which is a positive reflection of the quality of the care network for the diagnosis and treatment of breast cancer. We also observed inequalities in access to health services related to age, region of residence, stage of the disease, race, and origin of referral to the health service.

## 1. Introduction

Breast cancer incidence has seen the greatest increase amongst all types of tumors over the last decade, and is considered a major public health problem worldwide. It is the most common malignant cancer in women and was responsible for the deaths of 521,907 women worldwide and 16,412 women in Brazil in 2012. There were approximately 59,700 new breast cancer cases in Brazil in 2019 [1,2]. 

Waiting time for the commencement of cancer treatment after diagnosis is used to assess the accessibility and quality of health services. There are several factors associated with delayed access to health services such as bureaucratization, a lack of human resources, the unequal geographical distribution of health services, education level, age, ethnicity, and type of treatment financing [3,4,5,6]. These must be taken into account when considering the treatment of breast cancer, as delayed access to health services, and thus, delayed treatment, may influence prognosis and survival [7]. 

In Brazil, Law 12.732/2012 established that treatment waiting time should not exceed 60 days after diagnosis [8]. In this regard, a national study carried out in different regions of the country warned of unequal access to health services and consequent delays in treatment initiation [4]. Similar results have been found in both developed and developing countries [9].

However, community access to health care has widened, mainly due to the creation and constant improvement of the Brazilian Unified Health System (UHS, called *Sistema Único de Saúde* [SUS] in Portuguese), created in 1990 through law 8080/1190 [10]. The system is maintained by the Federal Government and provides the whole country with basic treatment and services of low, medium, and high levels of complexity [10,11]. 

On the other hand, since it is relatively new, this system has barriers to ensure universal and equal coverage, i.e., to comply with its own principles, which makes epidemiological studies regarding existing inequalities crucial to improve both the system and the lives of people who count on it.

Nevertheless, it must also be considered that, during the period studied, the UHS made several advances in health care through the creation and implementation of public policies for cancer care, such as the National Policy on Cancer Care: Promotion, Prevention, Diagnosis, Treatment, Rehabilitation, and Palliative Care [12]. Established in 2005, its aim was to create a cancer care network, increasing health coverage and reducing cancer morbidity and mortality. It was only after the creation of this policy that there was a requirement to establish the Hospital Registry of Cancer (HRC), which had been nonexistent in oncological care institutions up to that moment [13].

These services are integrated with other primary care services, and work through referral and counter-referral to ensure the continuity of health care. However, poor geographical distribution, as well as the high number of philanthropic and private institutions that provide services complementary to the UHS, result in the generation of different referral sources, as well as influencing the waiting time for cancer treatment.

Perhaps as a solution, or a way to work around this problem, the Ministry of Health created an information system known as “*SISMAMA*” in 2009, which monitors early breast cancer detection [14,15]. SISMAMA information is initiated in the Basic Health Unit, where the health care professional identifies those requiring a mammography and refers them for examination in a radiological unit. When a tumor is suspected, the patient must undergo diagnostic investigation procedures that usually occur in medium and high complexity units where HRC is accessible. Diagnoses can be made at the same institution where treatment will be performed, depending on the original referral institution and other referral services available in the patient’s state of residence.

All things considered, variables and factors associated with delays in breast cancer treatment after diagnosis are of great importance for both women diagnosed with the disease and its progression and mortality. Thus, the aim of this study was to analyze which factors were associated with the waiting time between the diagnosis and treatment of breast cancer in women in Brazil between 1998 and 2012.

## 2. Materials and Methods 

### 2.1. Study Design

This is a cross-sectional study based on secondary data from Brazilian women diagnosed with breast cancer between 1998 and 2012.

### 2.2. Participants 

Brazil is a country with a higher territorial extension compared to the continental dimension; it is divided into five administrative regions: North, Northeast, Midwest, Southeast, and South. These regions differ with respect to their cultures and socioeconomic development. 

During the study period, 151,931 women were diagnosed with breast cancer in these different regions. We did not include male cases and nonanalytical data, which were insufficient to address questions regarding the health care provided, because they underwent treatment or follow-up in other hospitals (Figure 1).

### 2.3. Source and Data Extraction

Data were extracted from the HRC, a web-based system developed by the National Cancer Institute (NCA; INCA in Portuguese) available at the following URL: https://irhc.inca.gov.br/RHCNet/ [16]. 

The HRC is responsible for the collection, storage, processing, analysis, and dissemination of information regarding patients attending a hospital unit with a confirmed diagnosis of cancer. The information collected by the HRC reflects the standard of care provided, and is important for assessing the quality of health care offered to cancer patients [16]. This is also useful for research purposes, and provides a foundation for new public policies that aim to improve the Unified Health System. 

In brief, the HRC registers cancer cases, classifying them into two distinct categories: analytical cases, whose planning and treatment are carried out in the registering hospital, and which are priority cases according to the HRC; and nonanalytical cases, which arrive at the registry hospital having already been treated, do not undergo recommended treatment, are diagnosed through necropsy, or for whom therapy is not possible. More extensive information was collected for analytical cases, whereas only the identification of the patient and the tumor was possible for nonanalytical cases [17].

More resources are devoted to analytical cases, including follow-up over the subsequent few years, as well as specific and detailed analyses which are carried out according to the information available for this set of patients [17].

Data management was performed using Microsoft Excel version 2013 (Microsoft Corporation, Albuquerque, NM, USA).

Clinical and sociodemographic information such as age, education level, marital status, region of residence, race, and routing source were collected. The routing source corresponds to hospitals that treat cancer patients, acting as reference units in the health system, and receiving only patients referred by other health units of lower complexity who had already been diagnosed with cancer, or who need diagnostic confirmation. This item serves to assess the patients’ origins and how the referral system works with the primary and secondary health network. In some cases, the patient arrives at the service of their own accord; it must therefore be recorded whether the patient spontaneously sought treatment, or did so on the advice of friends, relatives, or other patients, except if the friend, family member, or other relative is a physician [17]. The status of entry into the service, clinical staging at diagnosis, family history of breast cancer, date of diagnosis, first consultation, and treatment start date were also recorded.

Upon hospital admission, patients were grouped accordingly: Group I (first consultation without diagnosis) corresponded to women that were referred to the hospital with a suspected breast cancer but without formal diagnosis. Therefore, the date of the first visit to the hospital preceded the date of the diagnosis. Group II (first consultation with prior diagnosis), representing most of the women in the study, comprised women who had already received a diagnosis from another health-care facility and not from the hospital in which they received treatment. Group III (first consultation with prior diagnosis and treatment) corresponded to a smaller proportion of the patients because, in these cases, the patients received diagnosis and treatment at another hospital and continued treatment at a later time at the hospital in which they were registered with the HRC [4].

The time between diagnosis and the first medical visit was calculated only for patients in Group II, because patients in Group III had already been treated and patients in Group I had no diagnosis previous to their first hospital consultation [4]. 

All groups (I, II, and III) were included in the analysis regarding the time between the first medical visit and start of treatment. Only Groups I and II were included in the analysis of the time between diagnosis and start of treatment, because the patients in Group III had already been treated. Finally, only Group I was included in the analysis of the time between the first medical appointment and diagnosis [4].

### 2.4. Ethical Issues

This study was performed with secondary data available in a public system where it was not possible to identify patients, so informed consent was not required. The Research Ethics Committee of *Faculdade de Juazeiro do Norte* approved this study (approval number: 1.179.211/2015).

### 2.5. Statistical Analysis

Qualitative variables were described using absolute and relative frequencies. Quantitative variables, given the nonnormality of the quantitative data using the Shapiro–Wilk test, were presented based on the median and 95% confidence intervals (CI), respectively. 

Logistic regression of the waiting time between diagnosis and treatment was adjusted according to the clinical and epidemiologic characteristics. The stepwise backward method was used for the addition and removal of variables in the model. The significance levels for the removal and addition of variables in the models were *p* ≤ 0.20 and *p* > 0.05, respectively. 

The statistical software used was Stata version 11.0 (StataCorp, LLC, College Station, TX, USA), and the significance level was set at α = 0.05.

## 3. Results

The study included 151,931 women with breast cancer. The majority of these came to the service without diagnosis or treatment (42.6%), were referred by the UHS (74.7%), had a family history of for breast cancer (47.5%), were at stage II or III at the time of diagnosis (72.8%), and waited ≤ 60 days before starting treatment (81.6%) (Table 1).

The highest concentrations of women diagnosed with breast cancer were in the Southeast (30.9%) and Northeast (31.3%) regions. They were married (53.1%), educated to an elementary school level (60.0%), and were between 50 and 69 years of age (46.24%) at the time of diagnosis (Table 2).

The results from comparisons of waiting times of ≥60 days between diagnosis and treatment in women with breast cancer and epidemiologic characteristics that were statistically significant are detailed in Table 3. White women had a 5% higher chance of waiting 60 days or more to start treatment after diagnosis (OR, 0.95; 95% CI, 0.94–0.97; *p* = 0.001) compared to nonwhite women. With respect to education levels, only those with elementary school education were more likely to experience delays in treatment (OR, 1.15; 95% CI, 1.10–1.21; *p* < 0.001). Those who sought the service of their own accord or had a private health plan were less likely to experience delays (OR, 0.35; 95% CI, 0.32–0.39; *p* = 0.001 and OR, 0.93; 95% CI, 0.90–0.97; *p* < 0.001) compared to those referred by the UHS. The likelihood of delay increased proportionally with age (*p* < 0.001).

## 4. Discussion

The main findings of this study were that there was a greater chance of waiting longer than 60 days to start treatment for breast cancer for women of older age, nonwhite race, referred by the Public Health System (UHS), diagnosed as stage I or II, and having a low education level, mainly when a biopsy had already been performed. Nonetheless, we observed that only a few women with breast cancer in Brazil waited longer than 60 days to start treatment after diagnosis during the studied period (Table 1). 

We also observed that older age and lower education level were related to an increased risk of delay. This seems to be a common phenomenon, similar to that found in studies conducted in countries such as Tunisia [18], USA [19,20], Estonia [21], Pakistan [22,23], and Malaysia [24].

Regarding race, although there is a well-established relationship with delay between diagnosis and treatment of breast cancer, a study conducted in the USA showed that non-Hispanic/white race is more associated with delays in diagnosis than age, low educational level, or negative family history of breast cancer [25]. In New Jersey, a 5-year retrospective cohort study found that African-American women are more likely to experience delays in their treatment after diagnosis compared to white women [26]. The nonwhite Brazilian women in our study were also more likely to experience delayed treatment. 

Nonwhite women with a low education level residing in the North waited longer between diagnosis and treatment compared to the other groups. The Brazilian Northern Region is characterized by a greater territorial extension, poor socio-economic development, and physical barriers to health care that include the need for river transportation, inequalities in income, and longer distances to the centers of development in the Southeast region [6].

The major cancer treatment centers are located in the Southeastern region, where a significant proportion of women from other regions of the country are referred, which may be related to the longer waiting times in this region. Large territorial extension, poor distribution of income, as well as low socioeconomic development, are factors that represent barriers to health services [4,5].

Moreover, those referred by the UHS or diagnosed as stage I or II are more likely to experience delays compared to others [7]. From these results, we showed that waiting times were significantly unequal among the different regions of Brazil.

Regarding the origin of access, women who had a health plan or who had access by private means were less likely to experience delays greater than 60 days between diagnosis and treatment, which is in agreement with a study carried out in Mexico [3] and another prospective study carried out in Brazil [5]. In both studies, women who had access to health services through private or health plans received better and more efficient treatment, reflecting positively on the outcome of the disease.

Consequently, women entering the referral and counter-referral system of the public health system are at a disadvantage because they wait longer to start treatment compared to those sent by the private sector/health plan or of their own accord, which does not conform to the principle of equality that is the basis of the UHS [27].

Several public policies were created with the aim of increasing the coverage of screening for early diagnosis and the expansion of the care network for cancer treatment. This occurred due to the need to structure services in a regionalized and hierarchical way to guarantee an equitable distribution of resources among the population, and access to consultations and examinations for cancer diagnosis [1,12,28,29]. 

Contrary to findings in studies in other developing countries [9], where delays are common in both diagnosis and treatment, only a few women in our study experienced delays in diagnoses, i.e., a delay longer than that mandated by law. Although observed among only a few of the women in the study, these delays contribute to diagnoses in advanced stages, and thus, worse prognoses [30,31] and survival [7]. In order to reduce waiting times in Brazil, the law states that the maximum period between diagnosis and the start of treatment should be no more than 60 days (law number 12732/2012) [9].

The primary health service remains deficient and there are no guidelines or requests for subsidiary examinations by the health care providers who care for these patients. Delays in scheduling specialized consultations in referral centers are still considerable, and there are no secondary services to perform biopsies for suspected cases in an outpatient setting, which contributes to delays regarding diagnostic biopsies [4]. This may be the reason why there are still inequalities in access to health and overall services and, thus, delays in both diagnoses and treatment of breast cancer [32]. 

The limitations of this study are those that are inherent to a retrospective study, such as data loss, database input errors, and unavailable variables that could enhance the effectiveness of evaluating the quality of health services. 

## 5. Conclusions

Brazil is a country with great territorial extension and cultural and ethnic diversity, with inequalities in access to health services for women with breast cancer. There are several factors that could minimize delays in starting breast cancer treatment and reducing inequalities in access.

These issues, in a broader context, may also be related to income inequality, as reported by Figueiredo et al. [6]. Our results are useful for the formulation of public health policies, and demonstrate that the epidemiological profile and issues related to income inequality [33,34] affect the equality of access to health services [35].

In this sense, there is still a need to reduce bureaucracy in the health system and better distribute health services between regions, as well as creating health policies that are shaped according to the needs of each region.

## Figures and Tables

**Figure 1 ijerph-17-04030-f001:**
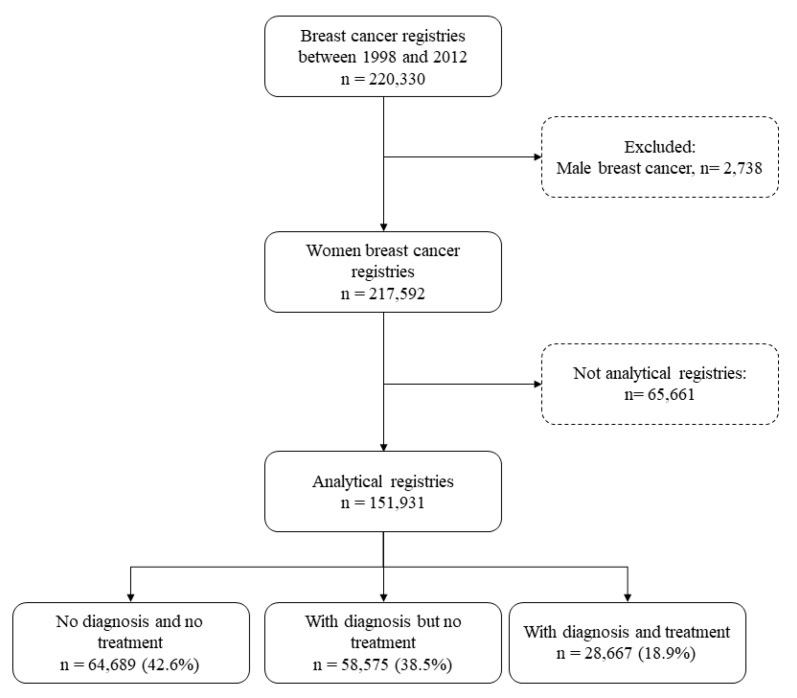
Data sorting procedure.

**Table 1 ijerph-17-04030-t001:** Clinical characteristics of women diagnosed with breast cancer in Brazil between 1998 and 2012.

Characteristic	*n*	%
Entry Status		
Group I—No Diagnosis/No Treatment (referred to hospital for diagnosis)	64,689	42.6
Group II—With Diagnosis/No Treatment (referred to hospital for treatment)	58,575	38.5
Group III—With Diagnosis/With Treatment (referred to hospital for continued treatment)	28,667	18.9
Stage		
I	19,972	17.6
II	45,954	40.6
III	36,549	32.2
IV	10,816	9.6
Routing Source		
UHS	100,168	74.7
Private health plan	28,152	21.0
Of own accord	5796	4.3
Family history of breast cancer		
No	5333	5.6
Yes	44,704	47.5
No information	44,142	46.9
Time between diagnosis and treatment		
≤60 days	100,984	81.6
>60 days	22,814	18.4

**Table 2 ijerph-17-04030-t002:** Epidemiological characteristics of women diagnosed with breast cancer in Brazil between 1998 and 2012.

Characteristic	*n*	%
Region		
North	6837	4.4
Northeast	48,712	31.3
Midwest	7237	4.7
Southeast	48,066	30.9
South	44,567	28.7
Marital status		
Single	31,589	23.4
Married	71,781	53.1
Widowed/Divorced	31,735	23.5
Education		
None	11,438	10.5
Elementary School	65,501	60.0
High School	21,410	19.6
Higher Education	10,723	9.9
Race		
Nonwhite	77,306	49.7
White	78,113	50.3
Age group (years)		
18–39	17,635	11.3
40–49	40,664	26.2
50–69	71,790	46.2
≥70	25,330	16.3

**Table 3 ijerph-17-04030-t003:** Comparison of waiting time (days) between diagnosis and treatment in women with breast cancer and epidemiological characteristics in Brazil between 1998 and 2012.

Characteristic	Diagnosis to Treatment ≥ 60 Days	Logistic Regression
	OR (95% CI)	*p*-Value
**Age group (years)**		
18–39	ref.	ref.
40–49	1.12 (1.06–1.18)	<0.001
50–69	1.28 (1.21–1.34)	<0.001
≥70	1.29 (1.21–1.36)	<0.001
Race		
Nonwhite	ref.	ref.
White	0.95 (0.94–0.97)	<0.001
Education level		
None	ref.	ref.
Elementary School	1.15 (1.10–1.21)	<0.001
High School	1.00 (0.94–1.07)	0.862
Higher Education	1.04 (0.97–1.13)	0.234
Region		
Northern	ref.	ref.
Northeast	0.53 (0.50–0.57)	<0.001
Midwest	0.35 (0.32–0.38)	<0.001
Southeast	0.71 (0.66–0.76)	<0.001
South	0.80 (0.75–0.85)	<0.001
Routing Source		
UHS	ref.	ref.
Private/health plan	0.93 (0.90–0.97)	0.001
Of own accord	0.35 (0.32–0.39)	<0.001
Stage		
I	ref.	ref.
II	0.96 (0.91–1.01)	0.076
III	0.73 (0.69–0.77)	<0.001
IV	0.70 (0.65–0.75)	<0.001

Ref., reference value; 95% CI, 95% Confidence Interval; OR, odds ratio.

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
