# Peer review of "Waiting Time between Breast Cancer Diagnosis and Treatment in Brazilian Women: An Analysis of Cases from 1998 to 2012"

_ijerph, 2020, doi:10.3390/ijerph17114030_

Round 1

Reviewer 1 Report

Brazilian Law No. 12.732 / 2012 established that breast cancer treatment must start within, no more than 60 days after diagnosis. The aim of this study was to investigate between 1998 and the law's enactment in 2012, factors that delayed breast cancer treatment. In this cross-sectional analysis of over 151,000 cases, the author's observed that only 18.7% experienced treatment delays. However the authors also observed that there were inequalities in access for individuals based on age, region of residence, and/or race / color and place of origin. The study is original and provides important information on Brazilian health care access. Findings are well presented and logical. The analysis is straight-forward. Conclusions are clear.

The study is valuable. Scientifically, I have only minor concerns. The manuscript, however, needs to be edited by someone who is a native English speaker. While the authors have excellent control of English, there are numerous grammatical flaws that if corrected, would significantly enhance the readability of the manuscript.

Minor concerns:

1.The word ethnicity is used when race is likely the more appropriate term. Hispanic is an ethnic designation. White, Black, Asian, Indiginous are race designations.

2.The authors show that White race (not ethnicity) was associated with increased wait-time. This observation should be more fully addressed.

3. Greater characterization of the North region would be helpful.

Author Response

Dear revisor #1. We thanks for your considerations and contributions. Please see the attachment.

Reviewer 2 Report

A valuable analysis of the timely access to breast cancer care in Brazil. However, the writing is difficult to follow and requires significant revision. It would be helpful to have a native English speaker review the manuscript for grammar and syntax. 

Examples of revisions required in the title and abstract:

1) The colon in the title is misplaced. Would either make it a comma, remove it and what follows it, or re-title, "Waiting time between breast cancer diagnosis and treatment in Brazilian women: an analysis of cases from 1998 to 2012."

2) First sentence of the abstract (line 12-13) is grammatically incorrect. Would re-word, "Brazilian law requires the treatment for breast cancer to begin within 60 days of diagnosis."

3) Line 14- should be "associated WITH."

4) Line 15- "regarding" should be replaced with "of."

5) Line 16- Would move the sentence "We performed secondary analysis" to after the sentence that ends  with "(INCA)." Would also add what you performed secondary analysis of.

6) How does "color" differ from "race" in your definition? If you are using these terms interchangeably, would remove "color" and just use "race," or perhaps substitute "ethnicity" for "color" if that is what you are referring to?

7) The factors you list in lines 20-21 do not match those you list in lines 25-26. For example, level of education and region of residence are in one list but not the other. Would make these consistent, or just list the factors in lines 20-21, and not list them again. You could end the last sentence after "... there is inequity in access to care."

8) Would break the last sentence of the abstract (lines 22-26) down into 2 sentences.

The introduction contains a lot of important background, but is difficult to understand in parts. The methods also require clarification-- you spend 3 paragraphs defining the three groups, but it is still not entirely clear. The way the 3 groups are listed in the tables and figures is confusing (i.e. "no diagnosis/no treatment." You have a lot of data here that could be further analyzed. For example, you could look at the association of this delay on stage at diagnosis and/or perform multivariable analysis to further elucidate factors contributing to delays. Has there been any change in the amount of delay over time (from earlier years of the study to more recent)?

Regarding the results, you could explain pertinent results further, rather than relying only on tables/figures. How was family history of breast cancer defined? If it is only first degree relatives, nearly 50% seems quite high. Table 3 contains a lot of important information-- a few clarification questions regarding the numbers:

1) How do you explain that no education has a similar level of delay as high school and higher education? This should be addressed in your discussion.

2) Many of the readers may not be familiar with the geography of Brazil. Would be helpful to add some background regarding the different regions to allow for interpretation of these results. You touch upon this in the discussion, but it should be presented earlier in the manuscript so readers know what to do with this data as it is presented in the results section.

3) How do you explain that higher stage is associated with shorter delay (if I am interpreting these data correctly)? This is counter to what has been published in many other studies.

Discussion/conclusion: Many of the findings in this study are consistent with what would be expected. It would make the manuscript more impactful to take some of what was learned in this analysis and make more directed/concrete suggestions for areas of future study and intervention.

Line 197- it is actually very well established that race/ethnicity are associated with delays.

Author Response

Dear revisor #2. We thanks for your considerations and contributions. Please see the attachment.

Reviewer 3 Report

Overall, the authors have done a decent job telling a story and identifying factors that contribute to disparities in length of time between diagnosis and treatment of breast cancer in a particular population of women.

Prior to resubmission, the authors should consider routing the manuscript through copyediting as there are instances of run-on sentences as well as a few grammatical and spelling errors (eg, "detailed analyzes" [line 111]; "Wom en Breast cancer" [Figure 1], lack of superscript registered symbols [line 113 and line 158], etc). Copyediting will also help better construct sentences for easier reading. With respect to the Results section, the tables need some slight revisions/formatting. Under the Entry Status category in Table 1, the phrase "from box 1" is used repeatedly but it is unclear what this is referring to (perhaps this should be "from Figure 1"?). In Table 2, the sub-headers under the Schooling category need to be reformatted to fit on 1 line respectfully for better readability. In Table 3, "OR" should be defined in the footer with the other abbreviations. On page 5, line 173-174 it reads "We observed that white women have a 5% chance of waiting more than 60 days...compared to nonwhites". The way this is currently stated may not clearly convey the intended point therefore, consider rewording to something like "White women have only a 5% higher chance of waiting more..." or "White women have a 95% lower chance of delays...". Rephrasing some of the statements in the Results section will help to increase clarity.

The Discussion section does a good job of comparing/contrasting the findings of the current study with those of others previously published in the literature. There were a few instances where the intent or purpose of statements were not entirely clear based on the rest of the manuscript. For example, line 220-223 seems to be a relative statement as UHS is disadvantageous relative to private health care, but the Introduction section of the current manuscript also suggests that UHS is better relative to no health care or the prior public health care system. Is the manuscript evaluating the efficacy of the UHS program or is the objective to evaluate factors associated with delays to treatment? Please clarify, as the former would require more background information and subsequent analyses.

Author Response

(The authors gave the same response as above.)

Round 2

Reviewer 2 Report

Thank you authors for replying to the suggestions made. While each specific comment has been addressed, the manuscript has not been reviewed by a native English speaker or copy editor, and would still benefit from English language review. There are still multiple grammatical errors, run-on sentences, and incorrect word choices which make the manuscript difficult to read. The discussion would benefit from improved organization.

I still find the correction in line 176 confusing in its wording.

Author Response

Dear reviewer,
We would like to thanks for your considerations. We changed the manuscript according to your suggestions and revised by a native.
Kind regards,
On behalf of the authors.
Dr. Francisco Winter,